# Neutrophil Extracellular Traps (NETs) in the Cerebrospinal Fluid Samples from Children and Adults with Central Nervous System Infections

**DOI:** 10.3390/cells9010043

**Published:** 2019-12-23

**Authors:** Daniel Appelgren, Helena Enocsson, Barbro H. Skogman, Marika Nordberg, Linda Perander, Dag Nyman, Clara Nyberg, Jasmin Knopf, Luis E. Muñoz, Christopher Sjöwall, Johanna Sjöwall

**Affiliations:** 1Division of Drug Research, Department of Medical and Health Sciences, Linköping University, SE-581 85 Linköping, Sweden; daniel.appelgren@liu.se; 2Division of Neuro and Inflammation Sciences, Department of Clinical and Experimental Medicine, Linköping University, SE-581 85 Linköping, Sweden; helena.enocsson@liu.se (H.E.); christopher.sjowall@liu.se (C.S.); 3Center for Clinical Research Dalarna-Uppsala University, Region Dalarna and Faculty of Medicine and Health Sciences, Örebro University, SE-702 81 Örebro, Sweden; barbro.hedinskogman@regiondalarna.se; 4Åland Central Hospital, Department of Infectious Diseases, AX-22 100 Mariehamn, Åland, Finland; marika.nordberg@ahs.ax (M.N.); linda.perander@ahs.ax (L.P.); clara.nyberg@aland.net (C.N.); 5Bimelix AB, AX-22 100 Mariehamn, Åland, Finland; dag.nyman@aland.net; 6Department of Internal Medicine 3-Rheumatology and Immunology, Universitätsklinikum Erlangen, Friedrich-Alexander-University Erlangen-Nürnberg (FAU), DE-91 054 Erlangen, Germany; jasmin.knopf@uk-erlangen.de (J.K.); luis.munoz@uk-erlangen.de (L.E.M.); 7Clinic of Infectious Diseases, Linköping University Hospital, SE-581 85 Linköping, Sweden; 8Department of Clinical and Experimental Medicine, Linköping University, SE-581 85 Linköping, Sweden

**Keywords:** neutrophil extracellular traps, cerebrospinal fluid, adults, children, central nervous system, infection, chemokines, cytokines, borrelia, virus

## Abstract

Neutrophils operate as part of the innate defence in the skin and may eliminate the *Borrelia* spirochaete via phagocytosis, oxidative bursts, and hydrolytic enzymes. However, their importance in Lyme neuroborreliosis (LNB) is unclear. Neutrophil extracellular trap (NET) formation, which is associated with the production of reactive oxygen species, involves the extrusion of the neutrophil DNA to form traps that incapacitate bacteria and immobilise viruses. Meanwhile, NET formation has recently been studied in pneumococcal meningitis, the role of NETs in other central nervous system (CNS) infections has previously not been studied. Here, cerebrospinal fluid (CSF) samples from clinically well-characterised children (*N* = 111) and adults (*N* = 64) with LNB and other CNS infections were analysed for NETs (DNA/myeloperoxidase complexes) and elastase activity. NETs were detected more frequently in the children than the adults (*p* = 0.01). NET presence was associated with higher CSF levels of CXCL1 (*p* < 0.001), CXCL6 (*p* = 0.007), CXCL8 (*p* = 0.003), CXCL10 (*p* < 0.001), MMP-9 (*p* = 0.002), TNF (*p* = 0.02), IL-6 (*p* < 0.001), and IL-17A (*p* = 0.03). NETs were associated with fever (*p* = 0.002) and correlated with polynuclear pleocytosis (r_s_ = 0.53, *p* < 0.0001). We show that neutrophil activation and active NET formation occur in the CSF samples of children and adults with CNS infections, mainly caused by *Borrelia* and neurotropic viruses. The role of NETs in the early phase of viral/bacterial CNS infections warrants further investigation.

## 1. Introduction

Lyme neuroborreliosis (LNB), which is caused by a complex of Gram-negative spirochaetes called *Borrelia burgdorferi sensu lato* and transmitted to humans by the *Ixodes ricinus* tick, is the most serious manifestation of Lyme disease (LD) [1]. *Borrelia* constitutes a common cause of bacterial infection of the central nervous system (CNS) among children and adults in Sweden [2,3,4,5] and Finland [6]. According to the diagnostic criteria for LNB [7], mononuclear pleocytosis must be present in the cerebrospinal fluid (CSF), and this is an indicator of an active infection. Despite being a bacterial CNS infection, LNB is characterised by a distinct mononuclear predominance in the CSF. The reason for the relatively low proportion of polynuclear cells, such as neutrophils, not only in the CSF [8] but also in the skin of patients with *erythema migrans* [9]—the earliest manifestation of a borrelia infection—remains unclear. However, as the neutrophils may be present at an early stage prior to lumbar puncture or skin biopsy, they might be overlooked.

Neutrophils are an essential component of the innate immune system, in that they limit locally the pathogen load through phagocytosis and the production of hydrolytic enzymes and chemokines, thereby orchestrating the subsequent adaptive immune response against the bacteria [10]. Beyond this, neutrophils create extracellular traps (NETs), which consist of the neutrophil’s own DNA in the form of de-condensed chromatin, histones, peptides including neutrophil elastase (NE), and myeloperoxidase (MPO). Thus, they entrap extracellular pathogens, causing bacterial death, in a process that was first described in 2004 and termed ‘NETosis’ [11,12,13]. NETs are capable of binding both Gram-negative and Gram-positive bacteria, and their activities not only result in bacterial death but also prevent further spread of the infection and maintain homeostasis at the point of entry, even in unexpected body compartments [14,15,16]. NETs can also ensnare and immobilise viruses, thereby preventing viral entry into cells and spreading [17].

The activities and roles of NETs are mostly unknown in the context of LD and human CNS infections overall. Nevertheless, NETs have been detected in the skin samples of Borrelia-infected, tick-bitten mice and have been shown to entrap and kill the spirochaetes despite a tick saliva-mediated decrease in the production of reactive oxygen species by the neutrophils [18]. In addition, NETs were detected in the CSF in a porcine model of *Streptococcus suis* meningitis [19] and in a human study of acute bacterial meningitis caused by pneumococci, in which the NETs were shown to play a harmful role in hinder the clearance of bacteria from the rat brain. However, in the same study, NETs were not detected in the CSF samples of patients with LNB (*N* = 3) and acute viral meningitis (*N* = 4) [20].

Neutrophils and their activation products are important early during infection, and they most likely affect the course and outcome of many diseases. With LNB, children tend to seek medical care earlier than adults owing to their having higher frequencies of meningeal symptoms and facial nerve palsy [3,4,21]. Therefore, we systematically evaluated whether NETs could be detected (using two different assays) in the CSF samples of children and adults with LNB, as well as in CSF samples from other infections and disorders affecting the CNS. NETs play a complex role in inflammation, although in general they are more stable than cytokines/chemokines and are thus more suitable for analyses of innate immune activation, especially in the sensitive environment of the CNS [22]. 

## 2. Patients and Methods

### 2.1. Paediatric Patients

The clinical characteristics of the children are shown in Table 1. CSF samples from 111 well-characterised children (64 girls, 47 boys; median age, 10 years; interquartile range [IQR], 5–15 years) with definitive or possible LNB (*N* = 28), tick-borne encephalitis (TBE; *N* = 3), enteroviral meningitis (EVM; *N* = 7), other viral infection (OVI, *N* = 4), and other non-infectious disorders with neurological or CNS symptoms without pleocytosis (*N* = 69; as described in Table 1) were obtained by lumbar puncture prospectively during the period 2010–2014 as part of a multi-centre study evaluating children with suspected LNB in Sweden, as previously reported [23]. LNB was diagnosed according to European guidelines [7]. Definite LNB (*N* = 20) was defined as the presence of: (i) symptoms attributable to LNB, (ii) mononuclear pleocytosis in the CSF, and (iii) intrathecally produced anti-Borrelia antibodies. All the children in the ‘possible LNB’ group (*N* = 8) had symptoms that were attributable to LNB and pleocytosis in the CSF but had neither detectable intrathecally produced anti-Borrelia antibodies nor clinical signs or laboratory evidence of other infections. These children all responded well to antibiotic treatment. Definite and possible LNB (*N* = 28) were referred to collectively as ‘LNB patients’. All samples were drawn before the initiation of antibiotic treatment, and the CSF samples for evaluation of NETs were frozen at −70 °C until analysis. All the children were evaluated for intrathecal anti-Borrelia antibody production (*Borrelia burgdorferi*-specific IgG and/or IgM), using the flagella antigen-based, enzyme-linked immunosorbent assay (ELISA, for serum and CSF) in the IDEIA Lyme Neuroborreliosis Kit (Oxoid Ltd., Hampshire, UK) [24]. An index >0.3 was considered as positive, indicating intrathecal production of anti-Borrelia antibodies according to the manufacturer’s instructions. Pleocytosis in the CSF was defined as ≥5 × 10^6^ cells/L. The children were followed for assessment of recovery 2 months after admittance to hospital.

### 2.2. Adult Patients

The clinical characteristics of the adult patients are shown in Table 2. CSF samples from 64 well-characterised adults (36 women, 28 men; median age, 55 years; IQR, 43–63 years) with LNB (*N* = 32), unspecified LD (*N* = 6), TBE (*N* = 3), other viral meningitides (OVM, *N* = 2), and other non-infectious disorders with neurological or CNS symptoms (*N* = 21; as described in Table 2) were prospectively collected by lumbar puncture as part of the clinical routine for patients who were being evaluated for suspected tick-borne CNS infection at Åland Central Hospital (Mariehamn, Finland) [7]. Definite LNB (*N* = 29) was defined as the presence of: (i) symptoms attributable to LNB; (ii) mononuclear pleocytosis in the CSF; and (iii) intrathecally produced anti-Borrelia antibodies. In addition, the level of CSF-CXCL13 was high (>100 pg/mL) in all the patients, supporting the diagnosis [25]. Patients in the possible LNB group (*N* = 3) had symptoms attributable to LNB and mononuclear pleocytosis in the CSF, and they showed neither clinical signs nor laboratory evidence of other infections. Increased levels of CSF-CXCL13 supported the diagnosis. The patients with definite and possible LNB ( *N* = 31) were collectively referred to as the ‘LNB patients’. Unspecified LD is a common manifestation in the highly Borrelia-endemic Åland Islands (personal communication, the Åland Borrelia Research Group, Åland Islands), where a large proportion of the population has anti-Borrelia antibodies in the serum and in the CSF due to previous exposure. Unspecified LD is defined as the presence of: (i) new symptoms suggestive of LD (headache, radiating pain, myalgia, arthralgia and fatigue); (ii) significantly increased anti-Borrelia antibody levels in the serum; (iii) a clear effect of antibiotic treatment for *B. burgdorferi* on current symptoms; (iv) a lack of pleocytosis and no increase in CSF-CXCL13; and (v) neither clinical signs nor laboratory evidence of other infections. Pleocytosis in the CSF was defined as ≥5 × 10^6^ cells/L. The CSF samples from adults were handled in the same manner as those from the children, i.e., frozen at −70 °C until analysis. CSF samples were analysed for anti-Borrelia IgG antibodies using the recomBead Luminex-based assay (Mikrogen, Germany). An index >0.3 was considered as positive, indicating intrathecal production of anti-Borrelia antibodies, in line with the manufacturer’s instructions. The adult patients were followed for the assessment of recovery 3 weeks and 6 months after admittance to hospital. 

### 2.3. NET Remnant Assay (NETs)

Phorbol-12-myristate-13-acetate (PMA; Sigma-Aldrich, St Louis, MO, USA)-induced NETs were used to create the standard curve for the ELISA. To induce NETs, neutrophils were isolated through Percoll density centrifugation, as described previously [26]. The neutrophils were re-suspended in culture media [RPMI 1640 plus 2% foetal bovine serum (FBS)], seeded onto a 12-well plate at 1.5 × 10^6^ cells per well, and stimulated with 20 nM PMA for 3 h. The wells were then washed twice with culture medium before incubation with 20 U/mL of the restriction enzyme *Alu*I (New England Biolabs Inc., Ipswich, MA, USA) at 37 °C for 20 min to cleave the NETs. The samples were centrifuged at 300× *g* for 5 min and NET-rich supernatants were pooled, aliquoted, and stored at −80 °C. 

The NET remnant ELISA was performed as previously described [27], although with PBS plus 0.05% Tween 20 used as the blocking and washing solution. A 96-well Nunc MediSorp immunoplate was coated with a monoclonal mouse anti-human MPO antibody (DAKO, Carpinteria, CA, USA) at 4 °C overnight. Blocking solution was added for 1 h at room temperature before incubation with standards (PMA-induced NETs), CSF samples, and a peroxidase-labelled anti-DNA antibody (detection antibody of the Human Cell Death Detection ELISA^PLUS^; Roche Diagnostics GmbH, Mannheim, Germany) for 2 h. Standards and samples were run in duplicate. For the standard curve, an 8-point dilution series with 2-fold dilutions of the PMA-induced NETs was used, and two plasma samples (high and low values) served as positive controls. After incubation, a substrate for peroxidase (ABTS; Roche Diagnostics) was added for 40 min before the plate was read at 405 nm in a VersaMax ELISA microplate reader (Molecular Devices, Sunnyvale, CA, USA). All the samples were interpolated from the standard curve using the sigmoidal 4-parameter logistic regression equation and reported as arbitrary units (a.u.). Non-detectable samples were given the value of 0 a.u. Analysis of NETs in the 175 CSF samples, which originated from the 111 children and 64 adults, was performed at the Division of Drug Research, Linköping University (Linköping, Sweden) [28].

### 2.4. Neutrophil Elastase (NE) Activity Assay

To confirm NET formation, 28 representative CSF samples (from 26 children and 2 adults) were also analysed for NE activity at the Friedrich-Alexander-Universität Erlangen-Nürnberg (Erlangen, Germany). For measurement of NE activity in the CSF, 100 µL of CSF were added to 100 µL PBS plus 25 µL of 1 M fluorogenic substrate MeOSuc-AAPV-AMC (sc-201163; Santa Cruz Biotechnology) plus 25 µL of 3.3 mM sivelestat (S7198; Sigma-Aldrich) or PBS in black 96-well plates (137101; ThermoFischer Scientific), as previously described [28]. Fluorescence readings were acquired on a TECAN Infinite 200 Pro using the filter set (excitation 360 nm, emission 465 nm) after 51 h of incubation at 37 °C. Assays were performed with technical duplicates and the results are expressed as mean fluorescence intensity (m.f.i.).

### 2.5. Cytokine and Chemokine Assays

CSF samples from 13 NET remnant-positive children (54% girls; mean age, 9 years; IQR, 5–13 years) and 25 age- and sex-matched NET remnant-negative children (52% girls; mean age, 9 years; IQR, 4.5–12 years) were analysed for chemokines and cytokines using two different magnetic Luminex^®^ assays (both from R&D Systems, Abingdon, UK). The Human Magnetic Luminex^®^ assay was used to measure the levels of CXCL1, CXCL6, CXCL8 (IL-8), CXCL10, and MMP-9, whereas a high-sensitivity assay (Luminex^®^ Performance Assay) was used for the quantification of TNF, IL-6, and IL-17A. The m.f.i. value of the respective analyte was obtained using FLEXMAP 3D (Luminex Inc., Austin, TX, USA), and the Bio-Plex Manager ver. 6.2 software (Bio-Rad Laboratories, Hercules, CA, USA) was used for data analysis. One of the 11 NET-positive samples was not available for the high-sensitivity assay. The samples were run in two different dilutions at a minimum, and the concordance between dilutions was checked carefully when concentrations had to be determined from different dilutions. Concentrations above the range of quantification were assigned the value of the expected concentration of the highest standard. An m.f.i. value below the lowest standard concentration was assigned a value of 0. The cytokines and chemokines were selected based on their associations with neutrophils and/or NETosis [29,30,31,32].

### 2.6. Statistics

Data were analysed using the GraphPad Prism ver. 8.01 software (GraphPad Software Inc., San Diego, CA, USA). Chi-square or Fisher’s exact test was used to compare discrete variables in 2 × 2 contingency tables. D’Agostino and Pearson omnibus normality tests were used to determine whether or not the data for continuous variables had a Gaussian distribution. For a non-Gaussian distribution of the data, the Mann–Whitney *U*-test was applied in comparisons between two groups with independent observations. The Kruskal–Wallis test was used for comparisons of more than two groups and if the Kruskal–Wallis test result was significant, Dunn’s *post hoc* test was performed. For correlation analyses, Spearman’s rank correlation coefficient (r_s_) was employed. In all the analyses, a *p*-value < 0.05 was considered statistically significant.

### 2.7. Ethics

The study protocol was approved by the Regional Ethics Review Board in Uppsala (Dnr 2010/106, children) and the local Ethics Committee of the Åland Health Care, Finland (2/2010, 1/2011, adults).

## 3. Results

### 3.1. CSF-NETs in Relation to Patient Diagnoses

NETs were detected in the CSF samples of significantly more children than adults (*p* = 0.01); in 14 (13%) children, whereof 7 (6%) with LNB, 4 (4%) with EVM, 2 (2%) with TBE, and 1 (1%) with idiopathic peripheral facial nerve palsy (IPFP) without pleocytosis (belonged to the OVI group). Only one adult patient, with meningitis and mononuclear pleocytosis caused by Varicella zoster virus (VZV), had detectable NETs (5 a.u.). The range of NETs levels were as follows: NET-positive patients with LNB, 2–18 a.u. (IQR, 3.2–12.7); NET-positive patients with EVM, 38–2643 a.u. (IQR, 56–2023); in the two NET-positive patients with TBE, 39–74 a.u.; and in the single NET-positive patient with IPFP, the level was 17 a.u. The proportions of NET-positive patients in the LNB (25%), TBE (67%), EVM (57%), and OVI (25%) groups did not differ significantly. Higher levels of NETs were observed among patients with fever (*p* = 0.002). The NETs levels correlated significantly with the total white blood cell counts in the CSF samples (r_s_ = 0.43, *p* < 0.001) (Figure 1A), and with polynuclear pleocytosis (r_s_ = 0.53, *p* < 0.0001) (Figure 1B) as well as mononuclear pleocytosis (r_s_ = 0.41, *p* < 0.0001) (Figure 1C), although not with age, gender or duration of symptoms. However, when the 14 NET-positive children were analysed separately, the levels of NETs were shown to correlate with polynuclear but not with mononuclear pleocytosis (r_s_ = 0.58, *p* = 0.03 and r_s_ = −0.10, *p* = 0.72, respectively) (Figure 1D,E). Children with EVM displayed a significantly shorter disease duration than the children with LNB (*p* = 0.03); and the number of polynuclear cells in CSF among paediatric patients showed an inverse significant correlation with the duration of neurological symptoms (data available for 77 of the 111 cases; r_s_ = −0.45, *p* < 0.0001).

### 3.2. CSF-NE Activity

NE activity was detected in 13/13 NET-positive children but was not detected in the NET-positive adult patient (overall concordance between methods: 93%). The levels of NETs and NE activity correlated significantly (r_s_ = 0.81, *p* < 0.0001) (Figure 1F). 

### 3.3. NETs in Relation to Clinical Outcome in Children

As we only found one NET-positive adult patient, we focused on evaluating NETs in relation to the disease prognosis for children (Table 1). The vast majority of the LNB cases (25/28; 89%) had recovered at the two-month follow-up. Two of the non-recovered children were NET-negative and one had low levels of NETs (2 a.u.). Accordingly, the occurrence of NETs in the CSF was not associated with recovery among the children with LNB (*p* = 0.72). All seven children with EVM were fully recovered at the two-month follow-up, four of whom were NET-positive. Among the three patients with TBE, one of the two patients who recovered was NET-positive, as was the patient who did not recover. Two of the four OVI cases had recovered at follow-up, whereas none had detectable NETs in the CSF. Only two patients in the group with other neurological and non-neurological disorders with CNS symptoms were NET-positive, and they had not recovered at follow-up.

### 3.4. Cytokines and Chemokines

All eight cytokines/chemokines were detected in at least one sample among the NET-positive (*N* = 13) and NET-negative (*N* = 24 or 25) CSF samples, respectively. The total number of samples within the range of quantification is shown in Table 3. The levels of all the cytokines/chemokines were significantly higher among children with NETs: IL-6 (*p* < 0.001), CXCL8 (*p* = 0.003), IL-17A (*p* = 0.03), TNF (*p* = 0.02), MMP-9 (*p* = 0.002), CXCL1 (*p* < 0.001), CXCL6 (*p* = 0.007) and CXCL10 (*p* < 0.001) (Figure 2A–H) and fever (*N* = 16); IL-6 (*p* < 0.001), CXCL8 (*p* = 0.001), IL-17A (*p* = 0.028), TNF (*p* < 0.001), MMP-9 (*p* = 0.009), CXCL1 (*p* = 0.002), CXCL6 (*p* = 0.002), and CXCL10 (*p* = 0.001). Furthermore, all the cytokines/chemokines, with the exception of IL-17A, were detected at significantly higher levels in the CSF samples from children with pleocytosis (*N* = 24) (*p* < 0.001). A strong and significant correlation was noted between the CSF levels of IL-6, CXCL8, TNF, MMP-9, CXCL1, CXCL6, CXCL10 and NETs, NE, and total CSF cell count, as well as mononuclear and polynuclear pleocytosis, respectively (Table 3). The levels of IL-6, CXCL8, TNF, MMP-9, CXCL1 and CXCL10 in the CSF samples obtained from the different patient groups, respectively, are shown in Figure 3A–H. All the cytokines, with the exception of IL-17A, were present at significantly higher levels in the patients with EVM compared to the patients with “other neurological and non-neurological disorders”. Furthermore, all the cytokines/chemokines, with the exceptions of CXCL1, CXCL6, and IL-17A, were detected at higher levels in the patients with LNB, as compared to the patients with “other neurological and non-neurological disorders”. CXCL6 was the only cytokine that was detected at significantly different levels when comparing the LNB group with the EVM patient group.

## 4. Discussion

In recent years, interest in NETs has shifted from their roles in innate immune defence to their pathogenic importance in several clinical conditions, such as autoimmune diseases, cancer, cardio-vascular morbidity, and most recently, in infections [14,33]. In the present study, we present, for the first time, evidence of NETs in the CSF of patients with LNB with meningeal inflammation, consisting of both mononuclear and polynuclear pleocytosis, as well as in patients with viral infections of the CNS. NETs are defined as complexes of DNA and MPO, although NE constitutes an additional NET component [34]. All but two of the CSF samples with detectable NETs displayed NE activity also, supporting the actual presence of NETs in the samples (concordance between methods: 93%). 

The levels of NETs showed strong variability between the different patient groups, ranging from 2 to 2000 a.u. Interestingly, the highest levels of NETs were detected in children with EVM and TBE, who also had the highest median levels of polynuclear cells. Children with EVM also had the highest median levels of NETs/neutrophil-associated cytokines and chemokines in their CSF samples (among the 38 patients in whom cytokines and chemokines were analysed). In addition, the levels of NETs correlated significantly with the numbers of polynuclear cells, i.e., including neutrophils, which were absent in the paediatric controls (Other disorders) and were overall present at lower levels in the adult patients with longer disease duration, which might explain the absence of NETs in these samples. The only adult who possessed detectable NETs had VZV meningitis with a symptom duration of several weeks and few polynuclear cells in the CSF.

The median pleocytosis values were comparable among children with LNB and EVM, although the proportion of polynuclear cells was higher in the latter group. This may be due to the fact that all the patients with EVM had meningeal signs and, consequently, a significantly shorter disease duration before admittance to hospital, as compared to the patients with LNB. In line with this, the numbers of polynuclear cells in the CSF were found to be inversely correlated with the duration of symptoms.

The finding of neutrophil activation in both LNB and EVM is intriguing, since these infections have similarities with respect to both clinical presentation and CSF cellular profile, which may include neutrophil predominance in the early phase [35]. Several studies have de facto shown the presence of neutrophil-activating cytokines and chemokines in the CSF samples of patients with LNB [36,37,38,39] and EVM [40,41], and the production of neutrophil chemo-attractants, such as CXCL1, CXCL8 and CXCL10 after *in vitro* stimulation of neuronal cells in response to *B. burgdorferi* spirochaetes [42]. Thus, there appears to be a prerequisite for neutrophil recruitment into the CNS in infections caused by *B. burgdorferi* and enterovirus. To clarify further the presence of neutrophils in these CNS infections, we analysed several potent neutrophil-activating cytokines and chemokines in the NET-positive samples. The results showed increased levels of almost all the measured cytokines/chemokines, and these levels strongly correlated with both poly- and mononuclear pleocytosis.

The chemokine CXCL10, also called interferon (IFN) γ-inducible protein 10 (IP-10), is produced by innate immune cells, such as neutrophils, and it stimulates chemotaxis of mononuclear cells into the CNS in response to meningitis/encephalitis caused by, for instance, enterovirus [43], *B. burgdorferi* [44], and TBE virus [45]. Furthermore, IP-10 has been proposed as a potential discriminator between bacterial and viral CNS infections [41]. Interestingly, CXCL10 was particularly increased in the EVM group, in which the proportion of polynuclear cells was highest. We also found especially increased levels of the fever-inducing cytokines TNF and IL-6 [46] in the EVM group, in which all the patients reported having fever. The chemokines CXCL1, CXCL6 and CXCL8 bind to the CXC-receptor 2 on neutrophils [47], activating them and enabling their entry into the CNS during infections [48]. The highest levels of these cytokines were, as expected, detected in the EVM patients. IL-17A, which is produced by Th17 cells, is a potent activator of neutrophils through stimulation of CXCL1 and CXCL8. Its pathogenic role in the CNS has been studied in LNB [36] and in other bacterial as well as viral CNS infections [49]. 

The paucity of NETs in the CSF of the adult patients in general, and in the adult LNB patients in particular, might be attributable to a delay between innate immune activation and CSF analysis through lumbar puncture, since the adult patients had longer duration of symptoms at admittance. While the reason for the modest neutrophil activation in LNB is unknown, it might involve an inhibitory effect of the spirochaetes on early pathogen recognition and elimination. Indeed, outer surface proteins (Osp), which are major virulence factors expressed on the surface of *B. burgdorferi* in the early phase of LD, are required for the spirochaetes to establish infection. OspB have been shown to inhibit the phagocytosis and oxidative burst of human neutrophils and may interfere with complement activation [50]. OspC is known to protect *Borrelia sp.* from phagocytosis by mononuclear phagocytes, and also to some extent by neutrophils [51].

Since extracellular traps are not produced exclusively by neutrophils, being also produced by mononuclear cells, termed monocyte or macrophage extracellular traps (METs) [52,53], we calculated the proportions of polynuclear and mononuclear cells in the NET-positive samples. It proved to be specifically the polynuclear cells, rather than the cause of the pleocytosis (i.e., the diagnosis), that correlated with the levels of NETs. Moreover, the NET levels and NE activity showed a high degree of correlation. Therefore, we conclude that the NET structures detected were primarily of neutrophil origin. Despite measuring the NET components with two different methods, we cannot affirm unambiguously that the presence of these components is exclusively the result of NET formation, which is always a limitation when one is not studying the actual cellular processes. To investigate this aspect further, cells from the CSF need to be monitored in detail.

We are unable to draw firm conclusions regarding the levels of NETs in relation to disease recovery in any of the patient groups, except in the paediatric LNB group, in which the occurrence of NETs at admission was shown not to be associated with poor clinical outcome at the two-month follow-up. The strengths of the present study are that it is hypothesis driven and has a systematic design, in addition to having well-characterised patients and concordant results. However, the uneven distribution of patients in the different groups and the heterogeneity within the group of other disorders are limitations that made comparisons of the results between patients more difficult.

## 5. Conclusions

We demonstrate the presence of NETs in the CSF samples collected from patients with LNB and viral CNS infections and show that NETs strongly correlate with the presence of pleocytosis and neutrophil-stimulating cytokines/chemokines in the CSF. Further studies of human neutrophils and their activation products are warranted, so as to elucidate their pathogenic and prognostic roles in acute bacterial and viral infections of the CNS.

## Figures and Tables

**Figure 1 cells-09-00043-f001:**
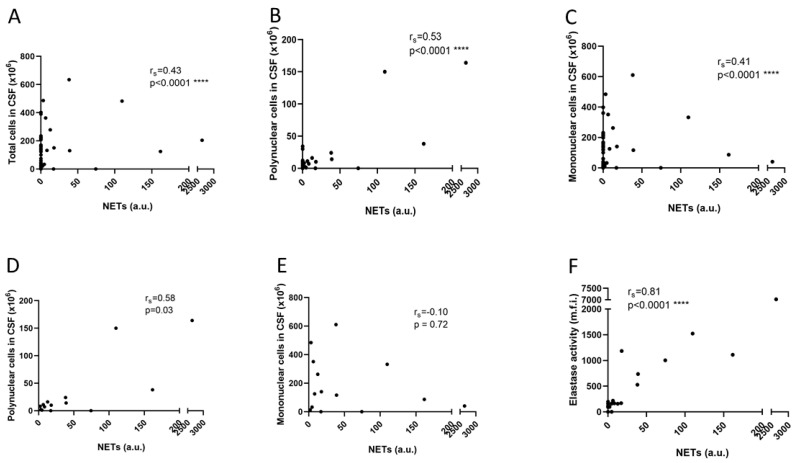
Correlations between cell counts in the CSF and NET remnants: (**A**–**C**) (*N* = 111 children) and (**D**–**E**) (*N* = 14 children). Correlation between NET remnants and elastase activity: (**F**) (*N* = 28; 13 NET-positive and 15 NET-negative cases).

**Figure 2 cells-09-00043-f002:**
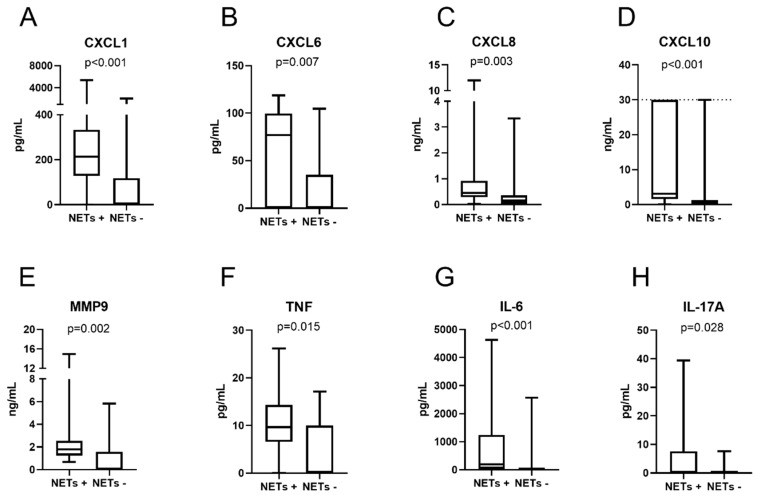
Concentrations of chemokines (**A**–**D**), MMP9 (**E**) and cytokines (**F**–**H**) in the CSF samples from the NET remnant-positive and NET remnant-negative children. Whiskers indicate the min-max values. The *p*-values are derived from the Mann–Whitney *U*-test. CXCL10 concentrations above the range of quantification (>30 ng/mL, *N* = 5) were assigned a value of 30 ng/mL (upper range of quantification is indicated by a dashed line). NETs: neutrophil extracellular traps.

**Figure 3 cells-09-00043-f003:**
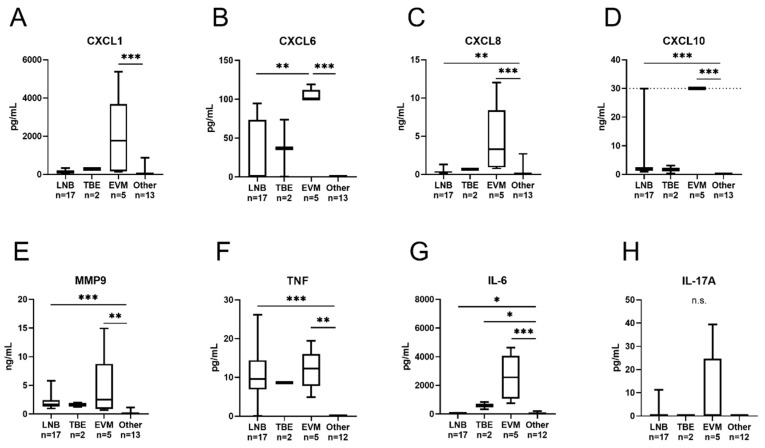
Concentrations of chemokines (**A**–**D**), MMP9 (**E**) and cytokines (**F**–**H**) in the CSF samples from the different patient groups. Whiskers indicate the min-max values. CXCL10 concentrations above the range of quantification (>30 ng/mL, *N* = 5) were assigned a value of 30 ng/mL (upper range of quantification is indicated by a dashed line). Only one child (rotavirus infection, no pleocytosis) belonged to the OVI group and is not included in the figure. LNB. Lyme neuroborreliosis; TBE, tick-borne encephalitis; EVM, enteroviral meningitis; Other, other non-infectious neurological and non-neurological disorders with CNS symptoms. * *p* < 0.05, ** *p* < 0.01, *** *p* < 0.001

**Table 1 cells-09-00043-t001:** Characteristics of paediatric patients.

On Admission		LNB (*N* = 28)	TBE (*N* = 3)	EVM (*N* = 7)	OVI (*N* = 4)	Other Disorder (*N* = 69)
Age	Median (range)	10 (3–15)	10 (3–15)	10 (4–15)	4 (1–14)	13 (0–19)
Sex	Female, *N* (%)	14 (50)	0	2 (30)	3 (75)	44 (64)
Duration of symptoms	<1 week, *N* (%)	16 (59)	1 (50)	5 (83)	1 (100)	9 (22)
	1–4 weeks, *N* (%)	9 (33)	1 (50)	1 (17)	0	11 (27)
	1–2 months, *N* (%)	1 (4)	0	0	0	5 (12)
	>2 months, *N* (%)	1 (4)	0	0	0	16 (39)
Clinical features	Facial nerve palsy, *N* (%)	22 (79)	0 (0)	0 (0)	1 (25)	17 (25)
	Meningeal symptoms ^¤^, *N* (%)	22 (79)	2 (67)	7 (100)	1(25)	50 (72)
	Fever >38 °C, *N* (%)	13 (46)	2 (67)	7 (100)	2 (50)	7 (10)
	Fatigue, *N* (%)	23 (82)	3 (100)	6 (86)	4 (100)	38 (55)
Laboratory findings	Pleocytosis *, median (range)	157 (20–486)	100 (0–130)	156 (20–634)	0 (0–6)	0 (0–74)
	CSF mono, median (range)	149 (8–484)	77 (0–116)	86 (16–610)	6	0 (0–40)
	CSF poly, median (range)	4 (0–30)	14 (0–28)	24 (4–164)	0	0 (0–34)
Recovery at follow-up:						
*2 months*	Yes, *N* (%)	25 (89)	2 (67)	7 (100)	2 (50)	45 (65)

CSF, cerebrospinal fluid; LNB, Lyme neuroborreliosis; LD, Lyme disease; TBE, tick-borne encephalitis; EVM, enteroviral meningitis; OVI, other viral infection; Other disorders (Demyelinating polyneuropathy, Idiopathic peripheral facial nerve palsy, Idiopathic intracranial hypertension, Epilepsy, Head trauma, Myasthenia, Infantile spasm, Autoimmune encephalitis, Guillan-Barré syndrome, Narcolepsy, Optic neuritis, Fatigue, Microcephaly, Mycoplasma infection, Voice hallucinations, Periodic fever aphthous stomatitis pharyngitis cervical adenitis (PFAPA), Papillary oedema, Multiple sclerosis). * ≥5 × 10^6^ cells /L; ^¤^ Meningeal symptoms: Headache, Neck pain, Neck stiffness.

**Table 2 cells-09-00043-t002:** Characteristics of adult patients.

On Admission		LNB (*N* = 32)	LD Unspec. (*N* = 6)	TBE (*N =* 3)	OVM (*N* = 2)	Other Disorders (*N* = 21)
Age	Median (range)	58 (18–82)	52 (30–80)	50 (46–57)	60 (50–69)	51 (24–88)
Sex	Female, *N* (%)	15 (47)	6 (100)	1 (33)	1 (50)	13 (62)
Duration of symptoms	<1 week, *N* (%)	4 (13)	0 (0)	0	0	1 (5)
	1–4 weeks, *N* (%)	7 (23)	1 (20)	3 (100)	1 (50)	1 (5)
	1–2 months, *N* (%)	10 (31)	1 (20)	0	0	1 (5)
	>2 months, *N* (%)	11 (34)	3 (60)	0	1 (50)	18 (86)
Clinical features	Facial nerve palsy, *N* (%)	5 (16)	1 (17)	0	0	0
	Meningeal symptoms ^¤^, *N* (%)	15 (47)	3 (50)	2 (67)	2 (100)	10 (48)
	Fever >38 °C, *N* (%)	3 (9)	0	3 (100)	1 (50)	3 (14)
	Fatigue, *N* (%)	17 (53)	5 (83)	3 (100)	1 (50)	9 (43)
	Radiating pain, *N* (%)	22 (69)	2 (33)	0	1 (50)	9 (43)
Laboratory findings	Pleocytosis *, median (range)	43 (6–390)	5 (4–5)	82 (51–131)	33 (18–47)	6 (0–292)
	CSF mono, median (range)	67 (5–355)	NA	62 (49–118)	26 (6–45)	54 (17–91) ^&^
	CSF poly, median (range)	3 (0–45)	NA	13 (2–20)	7 (2–12)	106 (10–202) ^&^
Recovery at follow-up:						
*3 weeks*	Yes, *N* (%)	12 (38)	3 (50)	0	1 (50)	4 (20)
*6 months*	Yes, *N* (%)	24 (77)	6 (100)	2 (67)	1 (50)	8 (42)

CSF, cerebrospinal fluid; LNB, Lyme neuroborreliosis; LD, Lyme disease; TBE, tick-borne encephalitis; OVM, other viral meningitis; Other disorders (Guillan-Barré syndrome, Spinal disk hernia, Trigeminal neuralgia, Sinusitis, Hypermobility syndrome, Depression, Chronic fatigue syndrome, Benign intracranial hypertension, Dementia, Recurrent iridocyclitis, Cerebral ischemia, Subarachnoidal haemorrhage, Chronic musculoskeletal pain, Meningeal inflammation of unknown origin). NA, not analysed. * ≥5 × 10^6^/L; ^&^ Based on 2 samples (cell count <10 × 10^6^ not diff in mono/poly); ^¤^ Meningeal symptoms: Headache, Neck pain, Neck stiffness.

**Table 3 cells-09-00043-t003:** Spearman’s correlation between cytokines, chemokines, cells and NETs in CSF.

		NETs (Remnants)		NETs (Elastase) ^†^		Cells (Total)		Polynuclear Cells		Mononuclear Cells	
Analyte	Quantified *	r_s_	*p*-Value	r_s_	*p*-Value	r_s_	*p*-Value	r_s_	*p*-Value	r_s_	*p*-Value
CXCL1	21/38	0.61	<0.001	0.76	<0.001	0.52	0.006	0.69	<0.001	0.48	0.02
CXCL6	14/38	0.52	0.007	0.65	0.002	0.72	<0.001	0.74	<0.001	0.67	<0.001
CXCL8	38/38	0.53	0.006	0.64	0.003	0.60	<0.001	0.65	<0.001	0.57	0.002
CXCL10	32/38	0.56	0.002	0.66	0.002	0.79	<0.001	0.81	<0.001	0.76	<0.001
MMP9	25/38	0.53	0.006	N/A	n.s.	0.70	<0.001	0.65	<0.001	0.68	<0.001
TNF	22/37	N/A	n.s.	N/A	n.s.	0.81	<0.001	0.65	<0.001	0.80	<0.001
IL-6	28/37	0.70	<0.001	0.81	<0.001	0.63	<0.001	0.73	<0.001	0.59	<0.001
IL-17A	4/37	N/A	n.s.	N/A	n.s.	N/A	n.s.	0.46	0.03	N/A	n.s.

* Samples within range of quantification (*N*)/total number of samples analysed (*N*). All samples outside the range of quantification were below the range of quantification, except for CXCL10; ^†^ Data on NETs by elastase assay was only available in 26 samples (CXCL1, CXCL6, CXCL8, CXCL10, MMP9) or 25 samples (TNF, IL-6 and IL-17A); *p*-values have been multiplied by the number of analytes (i.e. 8) to adjust for multiple testing. Grey shading indicates r_s_ ≥ 0.70. n.s. = non-significant, N/A = Not applicable.

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
