# Peer review of "Neutrophil Extracellular Traps (NETs) in the Cerebrospinal Fluid Samples from Children and Adults with Central Nervous System Infections"

_cells, 2019, doi:10.3390/cells9010043_

Round 1

Reviewer 1 Report

In this investigation, Appelgren et al examined cerebrospinal fluid (CSF) from 111 well-characterized children and 64 adults with LNB and other CNS infections was analyzed regarding NETs (DNA/myeloperoxidase complexes) and elastase activity. They found that NETs were detected in more children than adults. Presence of NETs was associated with higher CXCL1, CXCL6, CXCL8, CXCL10, MMP-9, TNF, IL-6, and IL-17A levels. Some cytokine levels are correlated well with NETosis and the significance of this correlation is not quite clear. Clinically, NETs associated with fever and correlated with polynuclear pleocytosis. Based on these findings, they conclude that CNS Borrelia infection and neurotropic viral infection induce neutrophil activation and active NET formation in the CSF of children and adults.  This is a descriptive human study to show the correlation between viral and bacterial infection and NETosis. Experiments were well performed, and data were analyzed appropriately.

Author Response

In this investigation, Appelgren et al examined cerebrospinal fluid (CSF) from 111 well-characterized children and 64 adults with LNB and other CNS infections was analyzed regarding NETs (DNA/myeloperoxidase complexes) and elastase activity. They found that NETs were detected in more children than adults. Presence of NETs was associated with higher CXCL1, CXCL6, CXCL8, CXCL10, MMP-9, TNF, IL-6, and IL-17A levels. Some cytokine levels are correlated well with NETosis and the significance of this correlation is not quite clear. Clinically, NETs associated with fever and correlated with polynuclear pleocytosis. Based on these findings, they conclude that CNS Borrelia infection and neurotropic viral infection induce neutrophil activation and active NET formation in the CSF of children and adults. This is a descriptive human study to show the correlation between viral and bacterial infection and NETosis. Experiments were well performed, and data were analyzed appropriately.

Response: We thank the reviewer for the overall positive feedback. In the revised version of the manuscript, further attempts have been made to clarify the correlations between certain cytokine levels and NETs (particularly in relation to the levels of CXCL10 in children with EVM).

Reviewer 2 Report

Cell Review.

Thank you for your work in investigating the prognostic significance of NETs in CSF from patients with CNS infections.  The authors found an association between NETs, inflammation cytokines and pleocytosis of PMN cells.

Please refer below for my comments and suggestions.

Methods and Materials:

NET remnant assay:

Please specify what blocking solution was used prior to incubation with standards, and CSF samples If this is a previously published technique, please include a citation of the literature. If this is a previously unpublished technique, then more details need to be provided here. How were the standards generated?  Please briefly include a description of how the “standards” were generated?  Standards imply that you generated defined concentrations of cfDNA by PMA activation and a dilution series were done to generate a range of absorbance.  Is this used as a positive control?   Please state clearly in your Methods and Materials sections the experiments performed to generate your PMA positive control.  This data should be presented clearly.  The 1 healthy male CSF donor served as control. I assume this was the biological negative control.  I am perplexed as to why n=1 would confidently give you enough power for any meaningful statistical comparison.  Please explain how the control subject was used in this context. How was the cut-off value determined? Please specify this in the Methods section.  Also state clearly what the cut-off was. One of the major drawbacks of this technique is that the results are entirely dependent on the presence of granulocytes in CSF. While I appreciate that the authors performed correlations between leukocyte counts and cfDNA concentrations, I recommend reporting the results as a ratio between cfDNA and granulocyte count.

Statistical Analysis

For continuous variables reported as median, please include the interquartile range.

Results

Line 179. Was the number of NET-positive CSF samples in each infective disease significantly different from? Line 185: Please include r values

Discussion

Line 243:  I would caution the authors to not overstate the results of this manuscript as the authors did not really show “active NET formation” in CSF.  Instead I would reword this statement as “presence” of NETs in CSF.

Lines 271 to 282s:  I think more explanation is needed as to why IL-10 is the only chemokine that has any significant correlation with NET formation in CSF.

Lines 297 to 298: A discussion regarding the limitations of the methodologies used to detect NETs is needed here.  cfDNA and NE activity, although NET markers, are non-specific to the cellular process of NETosis as they are both released during cell lysis and necrosis. 

The manuscript needs to be heavily edited for grammar and sentence structures. 

Author Response

Methods and Materials

NET remnant assay

Please specify what blocking solution was used prior to incubation with standards, and CSF samples If this is a previously published technique, please include a citation of the literature. If this is a previously unpublished technique, then more details need to be provided here. How were the standards generated?  Please briefly include a description of how the “standards” were generated?  Standards imply that you generated defined concentrations of cfDNA by PMA activation and a dilution series were done to generate a range of absorbance.  Is this used as a positive control?   Please state clearly in your Methods and Materials sections the experiments performed to generate your PMA positive control.  This data should be presented clearly.  The 1 healthy male CSF donor served as control. I assume this was the biological negative control.  I am perplexed as to why n=1 would confidently give you enough power for any meaningful statistical comparison.  Please explain how the control subject was used in this context. How was the cut-off value determined? Please specify this in the Methods section.  Also state clearly what the cut-off was. One of the major drawbacks of this technique is that the results are entirely dependent on the presence of granulocytes in CSF. While I appreciate that the authors performed correlations between leukocyte counts and cfDNA concentrations, I recommend reporting the results as a ratio between cfDNA and granulocyte count.

Response: We have now added a reference (ref. 26 Soderberg D, et al) that describes this ELISA in more detail, and we have also specified which buffer we used for blocking and washing.

Information on how the PMA-induced NETs were generated and collected has been included, as well as a reference that describes the neutrophil isolation procedure in detail. In addition, we provided a clarification regarding the standard curve, which was based on an 8-point dilution series (with 2-fold dilutions) of our generated NETs, and that samples interpolated from the standard curve are reported as arbitrary units (a.u.). As positive controls for the assay, we used one plasma sample from a patient with anti-neutrophil cytoplasmic autoantibody (ANCA)-associated vasculitis (high value) and one healthy blood donor (low value). As an internal negative control, CSF from a healthy male donor was included on each plate. However, we decided to exclude this information from the manuscript as we realised that it might be confusing, since it was not used to define NET-negative samples.

Concerning the detection levels, we have further clarified that all samples were interpolated from the standard curve using the sigmoidal 4-parameter logistic regression equation, and that non-detectable samples were assigned the value 0 a.u.

We appreciate the suggestion to report the results as a ratio of cfDNA to the polynuclear count. However, as 79 of the 111 samples (71%) were devoid of polynuclear cells, this yields division by zero. We believe that the correlation analyses (Figure 1a‒c) still facilitate the understanding of the extent to which the presence of polynuclear (or mononuclear) cells is related to the levels of MPO-DNA complexes. While these complexes might also have been released from mononuclear cells, as monocyte/macrophage extracellular traps also contain MPO and DNA, our data suggest a polynuclear cell origin in this case.

Statistical Analysis

For continuous variables reported as median, please include the interquartile range.

Response: We are not entirely sure as to which median values are being referred to here. Nevertheless, the interquartile range (IQR) has been added in the text of the Patients & Methods (Section 2.1, 2.2 and 2.5), as well as in the Results (Section 3.1), where the median age and NET values for the different patient groups are reported.

Results

Line 179. Was the number of NET-positive CSF samples in each infective disease significantly different from? Line 185: Please include r-values

Response: Line 179. There was a numerically larger proportion of children with EVM (57%) and TBE (67%) who were NET-positive compared to the LNB (25%) and the OVI (25%) groups. However, this did not meet statistical significance, probably due to the small groups. This information has been added to the revised manuscript.

Line 185. This point is well taken. The sentence was incorrect. In fact, higher levels of NETs were observed among patients with fever (binary variable). This information has been corrected in the revised manuscript.

Discussion

Line 243: I would caution the authors to not overstate the results of this manuscript as the authors did not really show “active NET formation” in CSF.  Instead I would reword this statement as “presence” of NETs in CSF.

Response: This is a good point, with which we agree, so we have removed “active formation” from the sentence. Please also observe our response to the comment below that cellular studies are needed to draw firm conclusions regarding the mechanisms of NET release.

Lines 271 to 282s: I think more explanation is needed as to why IL-10 is the only chemokine that has any significant correlation with NET formation in CSF.

Response: This is probably a misunderstanding. IL-10 was not measured. CXCL1, CXCL6, CXCL8, CXCL10 and IL-6 were all significantly correlated with NETs in the CSF samples. In particular, CXCL10 (also called IP-10) was present at very high levels in the EVM group, as compared to the patients with other infections (Figure 3), whereas also other cytokines/chemokines were higher among the children with EVM. This has been further highlighted in the Discussion section of the revised manuscript.

Lines 297 to 298: A discussion regarding the limitations of the methodologies used to detect NETs is needed here. cfDNA and NE activity, although NET markers, are non-specific to the cellular process of NETosis as they are both released during cell lysis and necrosis.

Response: We have elaborated on this in the Discussion by stating that we cannot conclude unambiguously that the release of DNA-MPO complexes is due to NET formation, that this is a limitation, and that studies on cells from the CSF are needed to elucidate this aspect further.

The manuscript needs to be heavily edited for grammar and sentence structures.

Response: The revised text has been comprehensively edited by a professional editor who is a native speaker of English and holds a PhD in Biomedicine.